# Effect of *Ligilactobacillus salivarius* and Other Natural Components against Anaerobic Periodontal Bacteria

**DOI:** 10.3390/molecules25194519

**Published:** 2020-10-02

**Authors:** Marzena Kucia, Ewa Wietrak, Mateusz Szymczak, Paweł Kowalczyk

**Affiliations:** 1R&D Depatrment Nutropharma LTD, Jedności 10A, 05-506 Lesznowola, Poland; marzena.kucia@nutropharma.pl (M.K.); ewietrak@nutropharma.pl (E.W.); 2Department of Molecular Virology, Institute of Microbiology, Faculty of Biology, University of Warsaw, Miecznikowa 1, 02-096 Warsaw, Poland; mszymczak@biol.uw.edu.pl; 3Department of Animal Nutrition, The Kielanowski Institute of Animal Physiology and Nutrition, Polish Academy of Sciences, Instytucka 3, 05-110 Jabłonna, Poland

**Keywords:** red complex bacteria, *Lactobacillus salivarius*, Salistat SGL03, periodontitis

## Abstract

In this present study, the bacteriostatic effect of Salistat SGL03 and the *Lactobacillus salivarius* strain contained in it was investigated in adults in in vivo and in vitro tests on selected red complex bacteria living in the subgingival plaque, inducing a disease called periodontitis, i.e., chronic periodontitis. Untreated periodontitis can lead to the destruction of the gums, root cementum, periodontium, and alveolar bone. Anaerobic bacteria, called periopathogens or periodontopathogens, play a key role in the etiopathogenesis of periodontitis. The most important periopathogens of the oral microbiota are: *Porphyromonas gingivalis*, *Tannerella forsythia*, *Treponema denticola* and others. Our hypothesis was verified by taking swabs of scrapings from the surface of the teeth of female hygienists (volunteers) on full and selective growth media for *L. salivarius*. The sizes of the zones of growth inhibition of periopathogens on the media were measured before (in vitro) and after consumption (in vivo) of Salistat SGL03, based on the disk diffusion method, which is one of the methods of testing antibiotic resistance and drug susceptibility of pathogenic microorganisms. Additionally, each of the periopathogens analyzed by the reduction inoculation method, was treated with *L. salivarius* contained in the SGL03 preparation and incubated together in Petri dishes. The bacteriostatic activity of SGL03 preparation in selected periopathogens was also analyzed using the minimum inhibition concentration (MIC) and minimum bactericidal concentration (MBC) tests. The obtained results suggest the possibility of using the Salistat SGL03 dietary supplement in the prophylaxis and support of the treatment of periodontitis—already treated as a civilization disease.

## 1. Introduction

Microbes live in harmony with the structures of the oral cavity. Tooth disease can be specified as a result of refraction or change of bacterial homeostasis. The oral epithelium is one of the physical mechanisms preventing the colonization of the oral cavity by hostile microorganisms and biofilm formation [1,2,3,4,5]. The biofilm of microorganisms themselves produce substances that inhibit other bacteria growth. The ability of some bacteria to adhere to the epithelium and teeth is one of the main factors determining the bacterial composition of the mouth [1,2,3,4,5,6]. The bacterial flora of the mouth has developed over the centuries to create a stable ecosystem. Microorganisms such as *Streptococcus* and *Lactobacillus* are the first colonizers and the main part of the natural bacterial flora of the oral cavity. According to the recent data there is over 700–1000 bacteria species that may colonize the human oral cavity like *Treponema, Lactobacillus, Propionibacterium*, *Selenomonas* and others [7,8,9,10,11,12,13,14,15]. They can cause apoplexy, common warts, shingles and smallpox [5,7,16,17,18].

Despite a rather high bacterial colonization, the oral cavity is actually relatively resistant to infection due to innate host defense mechanisms and biologically active components of the saliva. Moreover, mutual interaction between nonspecific and specific microorganisms present in the mouth helps to create and maintain homeostasis [16,19,20,21,22,23,24,25,26,27,28,29].

However, the presence of many pathogenic bacteria in the oral cavity, including red complex periopathogens, may cause disturbances in subgingival plaque homeostasis, which may lead to the development of severe and chronic periodontitis and various infections inducing inflammation in the oral cavity [2,5,17,18,27]. Red complex bacteria are very effective in blocking the body’s anti-inflammatory response to microbial attack. They inhibit monocyte chemotaxis by blocking the expression of adhesion molecules in intercellular reactions, blocking the action of interleukin-8. Currently, it is believed that the lack of an appropriate anti-inflammatory response is the main etiopathogenetic factor in the development of periodontitis—which affects the majority of adult populations worldwide [6,27].

Periodontal tissue infection causes inflammation that results in teeth moving. As a consequence, the pockets and then the roots of the teeth are exposed. This affliction leads to the destruction of structures keeping the tooth in the socket. The connection between the tooth and the surrounding tissues is weakened and, as a result, the teeth sway and even fall out. 

Microbiological and experimental studies in an animal model indicate that in the etiopathogenesis of chronic periodontitis pathogenic anaerobic non-sporulating bacteria, referred to as periodontopathogens or periopathogens, play a fundamental role [17,18,28,29,30,31,32,33,34,35,36,37].

The periodontium is then taken up by the many pathogenic bacteria of the red complex [5,7,8,17,18,19,28]. The dominant periopathogen is *P. gingivalis* [38,39,40,41,42,43,44,45]. In addition, significant etio-pathogenetic factors of periodontal disease in adults include: *T. forsythia*, *T. denticola*, *F. nucleatum*, *P. intermedia*, as well as the relative anaerobic *A. actinomycetemcomitans* [31,32,33,34,35,36,37,38,39,40,41,42,43,44]. *P. gingivalis* is clinically important bacterium species of the *Porphyromonas* genus, classified within the *Porphyromonadaceae* family found in over 40% of patients with periodontal disease [32]. *P. gingivalis* strains have fimbriae with numerous adhesins that ensure bacterial adhesion to periodontal tissues and allow coaggregation with other species, and also induce a pro-inflammatory cytokine response. At the same time, these strains produce exoenzymes—cysteine proteases (so-called gingipaines) with trypsin-like properties that cause tissue destruction [6,9,10,11,18,36,37,38,39,40,41,48]. Additionally, the final fermentation products of *P. gingivalis*, such as acetic, propionic and butyric acid, and volatile sulfur compounds, produced in large quantities, can have cytotoxic effects on host cells. These products disrupt tissue integrity and lead to local changes in the periodontal pocket microenvironment [6,36,41,46].

*P. gingivalis* activity similar to *T. denticola* and *T. forsythia* may occur during brushing, flossing, and chewing, as well as during dental procedures resulting in documented translocation to a variety of tissues including coronary arteries and liver [4]. *P. gingivalis* can be isolated from periodontal pockets, supragingival plaque, root canals, saliva, tongue, cheek mucosa as well as the tonsils [43,44,45,46,47,48,49,50]. Currently, based on the research by Socransky [44], specific groups of bacterial species (called complexes) are distinguished with particular importance in the etiopathogenesis of chronic periodontitis. The incidence of *P. gingivalis* is lower in people with a healthy periodontium compared to patients with periodontal disease. Most *P. gingivalis* strains have a polysaccharide coating that inhibits phagocytosis. The lipopolysaccharide (LPS) present in the cell wall-endotoxin, acts as an antigen and activates cytokines [30,36,37,50]. *P. gingivalis* forms together with the species: *T. forsythia* and *T. denticola* red complex that appears to be associated with chronic periodontitis in adults [34,35,45,47]. Synergistic with *P. gingivalis* are strains of the species *T. forsythia* of the genus *Tannerella*, also classified within the *Porphyromonadaceae* family. *T. forsythia* rods have an additional protective structure above the outer membrane—a surface layer of S formed by regularly arranged two protein subunits (MW 200 and 210 kDa). Experimental studies indicate that the S layer of *T. forsythia* strains can provide their adhesion to host cells as well as its invasiveness. Additionally, the final fermentation products of *T. forsythia*, such as acetic, propionic, and butyric acid, may have cytotoxic effects on host cells [48]. 

Microbiological studies show that there is a statistical relationship (positive co-relation) to the coexistence of oral spirochetes—*T. denticola* with *P. gingivalis* and *T. forsythia*, hence their belonging to the red complex [39]. Strains of the species *T. denticola* of the genus *Treponema* are classified within the *Spirochaetaceae* family, spiral bacteria characterized by active movement [39]. These spirochetes have the ability to migrate and penetrate into intact periodontal tissue. At the same time, the important factors of their virulence are produced enzymes: trypsin, chymotrypsin, alkaline phosphatase and esterase, enabling synergistic interaction of *T. denticola* strains with the other two species of the red complex [42,45]. 

Previous studies show that *Lactobacillus* spp. can modify the composition of the oral microflora by antagonist action against potentially pathogenic species [39,43,47,49,50]. For a more complete analysis of the etiopathogenesis of chronic periodontitis, it is therefore advisable to study the occurrence of *Lactobacillus* and periopathogen species in both moderate and severe forms of this disease. To date, there is a lack of data in the world literature analyzing this issue. In addition, in vitro studies on the effect of culture supernatants of selected periopathogens (*P. gingivalis*, *P. intermedia* and *A. actinomycetemcomitans*), as well as the culture of clinical Lactobacillus isolates on gingival fibroblasts, and evaluation of the expression of BTG2 genes encoding regulatory proteins with antiproliferative activity, preferentially produced in the G1 phase of the cycle, it is important for the analysis of the effect of metabolic products [33,34,35,36,37,38,39,40,41,42,43,44,45,46,47,48,49,50].

The periodontal disease associated with the bacterial plaque basic scheme prevention is based on the mechanical removal of dental deposits [2,29]. Specialized products that contain natural ingredients that favor long-term maintenance of the balance of oral microbiota are also helpful [2]. Research suggests that probiotic *L. salivarius* has the potential to modify the oral environment through antagonism to bacteria responsible for periodontal inflammation, including bacteria of the red complex [2,29]. *L. salivarius* SGL03 is a bacterium that occurs naturally in the healthy mouth and digestive system of humans. The strain has a sequenced genome, survives intestinal passage due to resistance to low pH and has no resistance to standard antibiotics; therefore, it meets the definition of a probiotic [12,13,24]. 

*L. salvarius* SGL03 has a fairly fast cell life cycle with division every 10 min at 37 °C. It reaches the highest number of cells after 8–9 h in the medium [27]. In an in vitro study by Pidutti [27], it was shown that *L. salivarius* produces L27 and L30 bacteriocins, which inhibit the proliferation of *S. pyogenes* within 4–6 h after being added to the medium [29]. *L. salivarius* SGL03 also inhibits the in vitro proliferation of such pathogenic bacteria as *S. mutans* and *S. sanguinis* [2]. Lactobacilluses have the immunomodulatory potential to reduce the production of various cytokines, such as (IL1b) [25]. The change in oral homeostasis towards the formation of pathogenic bacteria, related to the overgrowth of dental plaque, may cause bone damage as a result of increased host defense against attacks by pathogenic microorganisms [1]. Oral application of *L. salivarius* reduced the volatile sulfur compounds responsible for halitosis after 4 weeks of intervention [22].

Data suggest that the other natural and promising ingredient beneficial for oral health is lactoferrin. It is anti-inflammatory protein found naturally in human saliva. Lactoferrin achieves the highest concentration in body fluids and secretions e.g., in tears—2 mg/mL, in the vaginal secretion—0.15 mg/mL, while in the blood it is found at a concentration of only 0.2–05 μg/mL [13]. Some in vitro studies showed that this protein inhibits binding of *S. mutans, P. gingivalis, Prevotella intermedia* to the epithelium and tooth surface [1,2,14,15]. A pilot study on patients with moderate symptoms of periodontal disease showed that 4 weeks of topical use of lactoferrin as an oral hygiene supplement exerted a beneficial effect on gum health, including reduction in bleeding. Lactoferrin also may reduce bone resorption processes by inhibiting the inflammation factors like IL-8 [4]. Lactoferrin induces immunomodulatory, antiviral, antiparasitic, anticancer, effects and antibacterial and antifungal biofilm formation [1,3,15].

The healing properties of lemon essential oil have been used in herbal medicine for ages. Rosemary, *Rosmarinus officinalis*, is particularly popular in the Mediterranean region and is used primarily as a culinary herb, or a spice. Essential rosemary oil is extracted from the leaves of this plant. It is a natural disinfectant; therefore, it is often used as a natural component of a mouthwash. It also helps to remove bad breath and effectively removes bacteria from the mouth. It can prevent gingivitis, cavities, as well as plaque deposition and other destructive dental conditions. Rosemary and lemon oils showed antibacterial activity against aerobic bacteria causing dental caries and periodontal diseases such as: *S. mutans*, *S. sanguinis*, *S. pyogenes* and *P. gingivalis*. Lemon oil had a strong inhibiting effect against facultative anaerobes and anaerobes such as *Prevotella*, *Porphyromonas*, *Fusobacterium* isolated from patients with periodontal disease, inflammation of oral mucosa and perineal abscess. The highest efficiency of lemon oil in growth inhibition was achieved for *Staphylococcus aureus*—bacteria uses as control at a concentration of 7.5 mg/mL, lower inhibitory effect was showed for *Staphylococcus epidermidis*, *Enterococcus faecalis*, *Corynebacterium xerosis*. Lower effectiveness occurred in aforementioned cases, therefore to overcome the bacteria growth, a higher concentration of lemon oil should be used (above 20 mg/mL of solution), [12,16,21]. Natural oils have astringent and gentle refreshing effect, they also accelerate wound healing and inhibit the formation of biofilms on tooth enamel. It was shown that lemon oil has good efficacy in the control of Gram-bacteria in the treatment of mouth and throat diseases.

It is well known that the addition of prebiotic gluco-oligosaccharide (GOS) maintains the viability of the probiotic during passage through the digestive tract, being a source of nutrients for it [18,22]. 

Additionally, vitamin D, except for its effects on bone metabolism, has antimicrobial and anti-inflammatory properties, as well as healing and tissue regeneration potential. Available results suggest that the abnormal level of vitamin D may be connected with the development of periodontitis [14,23]. An in vitro study conducted by the Jagiellonian Innovation Center showed that dietary supplement Salistat SGL03, available on the Polish market inhibits growth of pathogens—S*treptococcus pyogenes*, *Streptococcus sanguinis*, *Streptococcus mutans* [4,7,17,18]. These are streptococci that start plaque formation and secrete extracellular polysaccharides (e.g., insoluble 1,3-α glucan) that stabilize plaque. 

Salistat SGL03, consisting of: *L. salivarius* SGL03 2 billion cfu, 50 mg lactoferrin, 0.5 mg lemon oil, 0.5 mg rosemary oil, 1000 mg gluco-oligosaccharides and 5 μg vitamin D is a biotechnologically advanced dental preparation, intended for the treatment of people with inflammatory changes in the oral cavity and periodontal diseases. On the basis of the characteristics of its individual components presented above, we hypothesized that the Salistat SGL03 dietary supplement, which is a proprietary blend of *L. salivarius* SGL03, lactoferrin, essential oils (lemon balm and rosemary), may significantly reduce the in vitro growth of selected pathogenic anaerobic bacteria associated with periodontitis and other oral diseases [24,25,26,27].

## 2. Materials and Methods

### 2.1. Microorganisms and Media

The reference bacterial strains of red complex (*P. gingivalis* ATCC 33277, *T. forsythia* ATCC 43037, *T. denticola* 35405 ATCC) were provided from (LGC Standards U.K.) and were used according to the recommendation of ISO 11133: 2014. Growth media like Bacteroides bile esculin (BBE) medium and Columbia agar for the red complex bacteria were from (BTL Company). Vials including dietary supplement Salistat SGL03 consisting of *L. salivarius* SGL03, lactoferrin, natural essential oils were kindly provided by Nutropharma LTD, Poland. A ready aqueous solution of active ingredients of Salistat SGL03 was used in a Petri dish with the pathogen culture and the same routine was applied on the 96-well plate. 

### 2.2. Determination of Minimum Inhibition Concentration (MIC) and Minimum Bactericidal Concentration (MBC)

The estimations were made using the minimum inhibition concentration (MIC) and minimum bactericidal concentration (MBC) method described previously by Kowalczyk [23,51]. The obtained results were statistically significant at the *p* < 0.05 level, measured by the Student’s *t*-test. 

### 2.3. Effect of Salistat SGL03 on Oral Microbiota Collected from Volunteers

Two experiments were performed independently: one in vitro and the other in vivo. In both experiments, a control group and a test group were separated. The material (bacterial inoculum) was taken from the oral cavity of female volunteers (hygienists)—control and experimental groups. The dental hygienist profession is generally female dominated, therefore, in these training groups only women applied for research, while men educated in this profession are very rare—hence the presence of only women in the analyzed research groups).

(a)The control group consisted of 10 generally healthy people (women aged 25–55 years) who had no history of systemic disease, had not received antibacterial drugs in the last four weeks, and had not smoked. The experimental group consisted of 10 people (10 women aged 25–55) diagnosed with moderate or severe chronic periodontitis on the basis of periodontal examination. These people did not use removable prosthetic restorations, did not receive periodontal treatment in the year preceding the examination, and did not use additional oral hygiene measures (dental floss, antiseptic liquids, irrigators). The mean disease duration was (16–20 months). All volunteers, after eating their first meal, did not brush their teeth for approximately 4 h before the experiment. Swabs were collected only from starch from the surface of the teeth after consuming a product that was registered in the Sanitary Inspection as a dietary supplement and is sold on the market. The participants of the study (volunteers) were informed about the course of the study and gave their consent. In the first stage, periodontal inoculum (s) were grown in Petri dishes with complete medium marked (P) and selective for Lactobacillus marked (L). Then 50 µl of an aqueous solution of Salistat SGL03 was placed in the cultured oral inoculum. The experiments were conducted to show the effect of *L. salivarius*, present in Salistat SGL03, on the formation of zones of inhibition of the growth of the oral microbiota in the complete (P) and selective (L) media. All collected Petri dishes were incubated overnight at 37 °C. The methodology used to induce the zones of growth inhibition is based on the disc method. diffusers (Kirby-Bauer). Therefore, a specific antibiogram was performed using the *L. salivarius* SGL03 strain contained in Salistat SGL03. In parallel, controls were also performed with all the ingredients of the formulation separately, such as lemon oil, rosemary oil and lactoferrin. After the plates were incubated with the seeded inoculum treated with spotted *L. salivarius* SGL03 contained in Salistat SGL03, the diameter of each of the zones of inhibition of microbial growth (including the diameter of the droplet) was measured and the reading recorded in mm. The measurement was made with a ruler or caliper (Figure 1).

As in the previous experiment, a second in vivo experiment was performed. The material was also collected from the oral cavity of twenty hygienists (volunteers—10 in each group).

(b)Additionally, the second inoculum was seeded on complete (P) and selective (L) media that were grown from the oral cavity after rinsing with Salistat SGL03 for approximately 30 s. This was done to show the effect of the product on inhibiting the growth of pathogenic bacteria. Similar to experiment (a), all seeded Petri dishes were incubated overnight at 37 °C (Figure 2).

In general, the study regarded the use of the preparation and the consent to collect the material from the teeth from the volunteers themselves, so the approval of the bioethics committee was not needed, as there was no invasive interference with the periodontal tissues. Pursuant to Polish law and EU law, consent for such tests is not required. However, the tests were performed in accordance with the legal acts presented below:

Act of 5 December 1996 on the profession of a doctor and dentist (Journal of Laws of 1997, No. 28, item 152) (Announcement of the Marshal of the Sejm of the Republic of Poland of February 22, 2019 on the publication of the uniform text of the Act on the professions of doctor and dentist, Journal of Laws 2019, item 537).

Act of September 6, 2001, Pharmaceutical Law, Journal of Laws 2001, No. 2001 No. 126, item 1381 (Announcement of the Marshal of the Sejm of the Republic of Poland of February 22, 2019 on the publication of the uniform text of the Act-Pharmaceutical Law, Journal of Laws 2019, item 499)

Act of September 6, 2001, Pharmaceutical Law (Journal of Laws of 2017, item 2211, as amended) and the Act of May 20, 2010 on medical devices (Journal of Laws of 2017, item 211) as amended)

Act of 1 July 2005 on the collection, storage and transplantation of cells, tissues and organs (Journal of Laws of 2005, No. 169, item 1411);

Act of 20 May 2010 on medical devices, Journal of Laws 2010, No. 2010 No. 107, item 679 (Announcement of the Marshal of the Sejm of the Republic of Poland of December 13, 2019 on the publication of the consolidated text of the Act on Medical Devices, Journal of Laws 2020, item 186)

Regulation of the Minister of Health of 2 May 2012 on Good Clinical Practice (Journal of Laws of 2012, item 489);

Regulation of the Minister of Health of 16 February 2016 on detailed requirements for planning, conducting, monitoring and documenting a clinical trial of a medical device, Journal of 2016 item 209

Recommendation (2006) four of the Committee of Ministers of the Council of Europe for member states on research on biological materials of human origin (Recommendation Rec (2006) four of the Committee of Ministers to member 13 states on research on biological materials of human origin), on 15 March 2006;

EU Directive 2004/23/EC of the European Parliament and of the Council of 31 March 2004 on setting standards of quality and safety for the donation, procurement, testing, processing, preservation, storage and distribution of human tissues and cells (OJL102, 7.4. 2004, p. 48).

Helsinki Declaration of the World Association of Physicians (WMA) Ethical Principles of Conducting Medical Research with Human Participation-October-2013.

### 2.4. Determination of Minimum Inhibition Concentration (MIC) and Minimum Bactericidal Concentration (MBC)

The estimations were made using the MIC and MBC method described previously elsewhere and by Kowalczyk [23,50]. The MIC determines what drug or compound concentration inhibits bacterial growth. The MBC determines what concentration of the drug has bactericidal activity. MICs were estimated using the microtiter plate method using sterile 48-well plates as previously described [21,50]. The material was also taken from oral cavity of 20 volunteers (hygienists) see (point b chapter 2.3). The bacterial inoculum was diluted in in 0.9% saline each, at a final concentration of 10 mM and 50 μL of each compound diluted to 1 mM was placed in the first line of rows of the plate. Next, to the other wells, 25 μL of sterile TSB medium was added and serial dilutions were performer from 10^−1^ to 10^−7^. Then, 200 μL of inoculated TSB medium containing resazurin indicator (0.02 mg mL^−1^) was added to all wells. TSB medium was inoculated to the concentration in ratio 1:100 (*v*:*v*) of the final suspension (~10^6^ cfu mL^−1^; about 0.5 McFarland) of all analyzed strains and plates were incubated at 37 °C for 24 h. A color change from blue to pink or yellowish and an increase in turbidity was considered positive, and the lowest concentration at which there was no visible colour change was assumed to be the MIC. Experiment was performed in three independent replicates. The MBC was estimated based on the measurement of dehydrogenase activity in cultures after a 24 h incubation. A 4 mL sample of a dense culture (~10^8^ cfu mL^−1^) that had been incubated for 24 h in TSB medium at 25 °C was added to a test tube, and tested compounds were added until the mixture reached a final concentration of 10−250 mg mL^−1^. The growth of the individual strains on the Salistat plates was incubated for 1 h at 30 °C. Then ~0.1 g of CaCO3 and 0.1 mL of a 3% TTC solution were added, mixed and the tubes were sealed with parafilm and incubated for 1 h at 30 °C in the dark. The lowest concentration at which there was no visible red color (due to formazan) was taken as the MBC. Then the MIC and MBC values were calculated.

### 2.5. Analysis of Red Complex Bacteria

The material containing the red complex strains was inoculated on Petri dishes by the reduction plating method in a given type of medium. Each bacterial strain complex of five periopathogens was suspended in 1 mL of sterile 0.9% saline (NaCl) to obtain a suspension with a density of 0.5 McFarland, which is approximately equal to 1–2 × 10^−8^ cfu/mL. The density of the suspension was determined by nephelometry using a colorimeter for 15 min. Excess suspension was removed by rotating and firmly pressing the swab or loop against the wall of the tube above the fluid level. The slurry was spread over the entire surface of Mueler–Hinton agar (MHA) three times, turning the plate each time. Within 15 min. the plates were placed in a laboratory incubator under aerobic conditions for 4 days at 37 °C for the efficient growth of the strains. On the fourth day, the cultured strains were inoculated with *L. salivarius* SGL03 present in Salistat SGL03 at 50 microliters per sterile 0.9% saline (NaCl) plate to obtain a 0.5 McFarland suspension which corresponds approximately to 1–2 × 10^−8^ cfu/mL. Then, the analyzed strains, along with the added *L. salivarius* present in the Salistat SGL03 preparation, were grown on the same plate for the next four days at 37 ° C. The results were interpreted (on the eighth day of the experiment) on the basis of the analysis of the degree of reduction in the growth of pathogens with the strains related to *L. salivarius* present in the preparation in Salistat SGL03.

### 2.6. Statistical Analysis

Data are presented as mean ± standard error (SE). A commercial Statistica for Windows packet (version 5.0) was used for the statistical analysis. Assumptions of normality and equal variance were tested before parametric analyses based on Student’s *t*-Test. Statistical significance was considered at *p* < 0.05.

## 3. Results

### 3.1. Determining Salistat SGL03 MIC and MBC on Pathogenic Bacteria

In our study, we analyzed the periopathogens of the red complex using standard microbiological methods, reduction culture, MIC, MBC, in the analysis of the oral microbiota after adding Salistat SGL03 to the strains. In all cases, it was observed that after treatment with all components contained in Salistat SGL03, the growth of pathogenic colonies was significantly inhibited compared to the control plate (Figure 3 and Figure 4). Data indicate that *L. salivarius* SGL03 present in Salistat SGL03 inhibits the growth of other Gram-positive microorganisms such as *Streptococcus* sp. (24). In all 48 analyzed wells (Figure 5) one to 12 with red complex bacteria, a visible color change was observed after addition of Salistat SGL03. Discoloration was obtained already at a dilution of 10^−2^, which corresponds to a MIC value of 0.25 mg mL^−1^ to 10^−4^, which corresponds to a MIC value of 0.0625 mg mL^−1^ (Figure 3), after the addition of resazurin (Figure 5)—used as the pH and redox indicator. Below pH 3.8, resazurin is yellow, and above 6.5 it is purple. As resazurin is reduced by living cells, it is used as a redox indicator in the cell in viability tests in aerobic and anaerobic bacteria. Overall, all bacterial strains showed high sensitivity to Salistat SGL03. The greatest sensitivity was observed for P. gingivalis. 10^−4^ than in strains of T.denticola and T. forsythia 10^−3^. The obtained results indicate that Salistat SGL03, and specifically its composition of ingredients, including L. salivarius, lactoferrin and natural oils, may have a bactericidal effect on pathogenic bacteria (Figure 6, panel A and B).

In all analyzed 96 wells (Figure 5A) from one to 12 with the bacteria of red complex a visible color change was observed after addition of Salistat SGL03. The discoloration was achieved already at dilution from 10^−2^ corresponding to MIC equal to 0.25 mg mL^−1^ to 10^−4^ corresponding to the MIC value of 0.0625 mg mL^−1^ (Figure 5) after adding of resazurin which induces microbial growth. 

A similar effect was observed for the concentrations in the MBC test (Figure 5B). Here, in all strains, a color change in the wells was observed already at a dilution of 10^−3^ for the analyzed products corresponding, to an MBC value of 0.125 mg∙mL^−1^. The order of arrangement of the strains was identical to that in the MIC. MBC values were one the similar level than the MIC. The obtained MIC and MBC values between individual strains after treatment with the analyzed Salistat SGL03 were statistically significant.

In general, all analyzed bacterial strains showed a high sensitivity to Salistat SGL03. The greatest sensitivity was observed for *P. gingivalis*, being 10^−4^ higher than for strains of *T. denticola* and *T. forsythia* 10(−3). The obtained results indicate that the components of Salistat SGL03, including *L. salivarius*, may have a bactericidal effect on pathogenic bacteria (Figure 6, panels A and B).

### 3.2. Results of Salistat SGL03 on Oral Microbiota Collected from Volunteers

The growth inhibition zone, visible after adding Salistat SGL03 to both types of media marked as P and L (Figure 7 panels A and B), showed the bacteriostatic effect of the Salistat SGL03 solution. Figure 7 panel A shows the oral cavity reduction cultures made after rinsing with Salistat SGL 03. In cases on control plates with complete (P) and selective medium for Lactobacillus (L), on which material from the oral cavity was taken, after rinsing with Salistat SGL03—(Figure 7 panel B) (see details in chapter 2.3 a and b materials and methods)—no growth of pathogenic bacteria was found.

In the case of using the complete growth medium marked as P (all microorganisms from the cultured oral cavity), the level of the zone of inhibition of growth (estimated in mm) on pathogenic bacteria was two times lower than in the selective medium for Lactobacillus strains marked as L (Figure 8). It can be indicated that the sensitivity of the bacterial microbiota after in vitro treatment with Salistat SGL03 strongly depends on the structure of the bacterial membrane and the LPS contained in it. All the obtained results were statistically significant at the level of *p* < 0.05 with the analyzed Student’s t-test.

## 4. Discussion

The obtained results suggest that active compounds of the tested dietary supplement Salistat SGL03 may interact with the red complex bacteria (Figure 1, Figure 2, Figure 3, Figure 4, Figure 5, Figure 6, Figure 7 and Figure 8). Earlier literature reports show that the active substances (*L. salivarius* SGL03, lactoferrin, essential oils) of Salistat SGL03 have antimicrobial activity against pathogenic red complex bacteria and cariogenic bacteria present in the human oral cavity.

The bacterial membranes of the tested strains showed different sensitivity to Salistat SGL03 based on the observed MIC and MBC values. Our preliminary results from using a product based on probiotics and other natural ingredients are very promising, especially in terms of a new approach in the treatment of periodontitis and maintenance therapy.

Chronic inflammation of the tissues surrounding and supporting the teeth leads to systemic complications. Periodontitis is associated with the progressive destruction of the bones of the alveolus and, without treatment, can lead to loosening and subsequent loss of teeth. This process can range from direct damage to tissues by bacterial products found in plaque, and indirect damage from bacterial stimulation of local and systemic inflammatory and immune responses. Recent studies also indicate a relationship between periodontitis and Alzheimer’s disease [52,53].

*L. salivarius* can significantly counteract tooth decay [2] and bacteria that cause periodontal disease—where anaerobic bacterial biofilms such as Ag exist. Actinomycetemcomitans, *P. gingivalis*, *T. forsythia*, *T. denticola*, *P. intermedia* [2]. The analyzed biofilms from both areas of the oral cavity contain the same bacterial composition [54].

A consequence of the subgingival colonization by bacteria of the structure surrounding the tooth that is bone, ligament (periodontium), gum and root cementum, are symptoms of oral cavity infection, as microbes can induce an inflammatory response that leads to redness, swelling of the gums, bleeding and eventually tissue degradation and through successive stages of loosening teeth—their recession [2]. In patients with periodontal disease, bacteria also colonize implants. Over time, the bacterial plaques on the surface of the teeth grow larger and descend below the gumline. Subsequently, bacteria and their products from the gingival pockets can pass through the blood vessels to the periphery and contribute to the development of systemic diseases. Therefore, the performance of MIC and MBC tests are an excellent “tool” for the analysis of these bacterial complexes. MIC and MBC are the basic indicators of the effectiveness of targeted antibiotic therapy and bacterial drug susceptibility testing in the chemotherapy of bacterial infections [55].

Currently, antibiotic resistance is becoming more common. The imbalance between the growing resistance of bacteria to the antibiotics used and the emergence of new, more potent drugs is so dynamic that the therapeutic options for treating bacterial infections have begun to decline. The 2015 American Dental Association guidelines for the non-surgical treatment of chronic periodontitis indicate that systemic antibacterial agents should be used with caution and the expected benefit to the patient is rather small [8,29,41,42,43,44,45,46,47,48,49,50,54,55]. The search for new substances that inhibit the growth of pathogenic bacteria in the oral cavity is currently a very high priority in scientific and clinical research. The mechanism by which Salistat SGL03 may be toxic to pathogenic red complex bacteria is the insertion of L. salistat SGL03 into the bacterial LPS cell membrane. The addition of lactoferrin, which also has a bactericidal effect manifested by binding to the cell membranes of LPS bacteria and essential oils (mainly their flavonoids and terpenes), which are toxic to bacteria, enhances the effect of this product on selected pathogens.

This suggests that the outer bacterial membrane, namely LPS, may react to Salistat SGL03 ingredients, that probably affect cell integrity and bacterial survival. Selected anaerobic strains used in our in vitro test were all sensitive to Salistat SGL03, as demonstrated by MICs. 

The use of the probiotic *L. salivarius* SGL03 in Salistat SGL03 after oral administration did not show any side effects related to the development of tooth decay. Microbiological analyses following oral inoculum removal (after analyzing the growth inhibition zones) showed better resistance to caries risk factors associated with reduced saliva flow and its buffering capacity. This is probably due to the positive effect of lactobacilli (reducing the acidity of plaque) [18,30,31], hydrogen peroxide and L27 and L30 bacteriocins secreted by *L. salivarius* to inhibit the growth of pathogenic carcinogenic strains: *S.pyogenes*, *S. sanguinis* and *S. mutans* and red complex periopathogens associated with periodontal diseases (*P. gingivalis*, *T. denticola*, *T. forsythia*) (own research). Allaker [2] tested in vitro the ability of eight probiotic Lactobacillus strains (*L. plantarum* 299 v, 931, *L. rhamnosus* GG, LB21, *L. paracasei* F19, *L. reuteri* ATCC PTA 5289, DSM 17,938 and L. acidophilus La5) which coagulate and inhibit the growth of *Streptococcus mutans*. All Lactobacillus strains showed co-aggregating activity and inhibited the growth of clinical *Streptococcus mutans*. Growth inhibition was strain specific and depended on both pH and cell density (29). The in vitro experiment also showed that the production of lactic acid in suspensions of dental plaque and probiotic lactic acid bacteria is strain dependent [44,50]. In this study, salivary pH did not change after consuming *L. salivarius* [56].

Short-term consumption of *L. rhamnosus* GG and *L. reuteri* for 2 weeks did not affect the production of lactic acid by the supragingival plaque [4,50]. However, after longer (6 weeks) consumption of *L. brevis* CD2, a significant reduction in the acidity of the plaque was found [50]. In addition, clinical evaluations have shown that the effects of probiotics vary with host susceptibility. In fact, levels of *Streptococcus mutans* in several participants increased after oral ingestion of *L. salivarius* strains. Moreover, the gluco-oligosaccharide present in the tested preparation decreased the ability of cariogenic strains to use it as an energy source for their metabolism. This is probably due to competition for adhesion sites, nutrients and growth factors between strains [46,49,50].

Lactoferrin, vitamin D, rosemary and lemon oil present in the preparation have an antiseptic effect, stimulating the immune system to temporarily reduce the secretion of interleukins (IL-1, 6 and 8) in the gingival fluid, reduce TNF-alpha necrotic factors (in people with chronic periodontitis) and accelerate mineralization of enamel. Additionally, rosemary oil inhibits the formation of biofilms on tooth enamel. On the other hand, lemon oil has an antagonistic effect on aerobic, relatively anaerobic bacteria and is isolated from patients with periodontal disease, periodontal abscess and oral mucositis. A pilot study involving people with moderate symptoms of periodontitis showed that the ingredients of Salistat SGL03, after topical application in oral hygiene, had an effect on gum health, resulting in a reduction in bleeding after 30 days [57].

These ingredients also exhibit antiseptic and astringent effects and may also inhibit the production of pro-inflammatory factors in the activity of osteoclasts (cells that dissolve bone tissue and affect bone resorption) [45,46,47,48,49,50].

However, lactobacilli producing hydrogen peroxide are associated with the maintenance of healthy intestinal microorganisms and the function of the immune system [48]. In the oral cavity, competition for glucose is likely to be high and therefore, *L. salivarius*, like *L. delbrueckii*, could compete for reduced carbon sources. It has been shown that during carbon deficiency *L. delbrueckii* autolyzes, releasing pyruvate oxidase and other intracellular components [33,34,35,36,37,38,39,40,41,42,43,44,45,47,48].

Therefore, adding bacteria such as *L.salivarius* to the mouth microbiome which can produce hydrogen peroxide can help maintain balance and eliminate other periopathogens from the biofilm that forms in the mouth. In addition, studies have shown that PPAR-, which is involved in anti-inflammatory responses and immune homeostasis, is activated by hydrogen peroxide and also produced by *L. crispatus* [48]. *L. salivarius*, similar to *L. delbrueckii* in an anaerobic or relatively anaerobic environment at the boundary between the gum gap and the tooth, produces hydrogen peroxide which directly affects the growth of *P. gingivalis*.

This aspect of hydrogen peroxide-producing bacteria is particularly important in the context of periodontitis, since most pathophysiological processes are caused by an excessive immune response to this process. Inhibition of *P. gingivalis* growth and its colonization in the oral cavity is critical as this bacterium may indirectly influence neoplastic diseases associated with pancreatic cancer and cardiovascular diseases [40,41,42,43,44,45,46,47,48,49,50,58]. The presence of the *L. salivarius* strain contained in Salistat SGL03 makes it potentially useful as a probiotic strain in the treatment and prevention of chronic periodontitis and colonization of *P. gingivalis* and other red complex periopathogens.

The results obtained suggest that Salistat SGL03 can be safely used without increasing the risk of caries, and that its oral administration may contribute to the control of dental plaque, healthy periodontium and the reduction in halitosis (unpleasant odor from the mouth) in healthy adults and patients undergoing periodontal therapy.

The Salistat SGL03 dietary supplement used is not subject to the regulations for medical devices or cosmetics that require such tests, e.g., for skin irritation or cytotoxicity or toxicity. The product is safe and used as a supplement to oral hygiene for the whole family as well as for pregnant and lactating women. It is intended for people who are preparing for or after orthodontic, prosthetic, surgical and endodontic procedures, after scaling and cleaning the roots of the teeth, with periodontitis. Dietary supplements are subject to the Food and Nutrition Safety Act, so they are part of the diet. However, further research into the characteristics of the probiotic strains and host response is needed to determine their appropriate use, doses and duration of treatment in dental, periodontological and ENT practice.

In the future, we intend to expand our research with a cytotoxicity test, skin sensitization test, acute systemic toxicity test, and a skin irritation test based on the international standard ISO 10993 which concerns the determination of the effect of medical devices on tissues. Except for EN ISO 10993-2: 2006 and EN ISO 10993-10: 2013, the remaining parts of EN ISO 10993 have the status of harmonized standards with Directive 93/42/EEC for medical devices and Directive 90/385/EEC for active medical devices for implantation. The provisions of these directives are implemented by the Act of 20 May 2010 on medical devices (Journal of Laws of 2019, item 175, as amended).

## 5. Conclusions

The obtained in vitro results indicate that components of dietary supplement Salistat SGL03 have potent antimicrobial effect on selected pathogenic bacteria responsible for periodontitis. Therefore, adding this product to your daily oral hygiene practice can be a good way to ensure proper oral health prophylaxis and supportive care, especially for patients with the first symptoms of gingivitis or early stage periodontitis. The obtained knowledge in this field may be of practical importance not only for the diagnosis of periodontal disease, but also may contribute to the optimization of disease therapy strategies. Therefore, undertaking the research presented in this paper justifies: the prevalence of chronic periodontitis in adults, difficulties in treatment, especially in severe forms, as well as very serious consequences of this disease, mainly tooth loss and generalized infections of the body. Epidemiological studies show that oral health is closely related to the body as a whole. Many studies to date have already characterized specific species of periopathogens and their virulence factors, as well as the most commonly co-occurring groups of bacterial species (called complexes) of particular importance in the etiopathogenesis of periodontitis. However, the actual contribution of periopathogens and the role of *Lactobacillus* bacteria in periodontal disease activity is still unknown, which is particularly important for a better understanding of the etiopathogenesis of this disease.

## Figures and Tables

**Figure 1 molecules-25-04519-f001:**
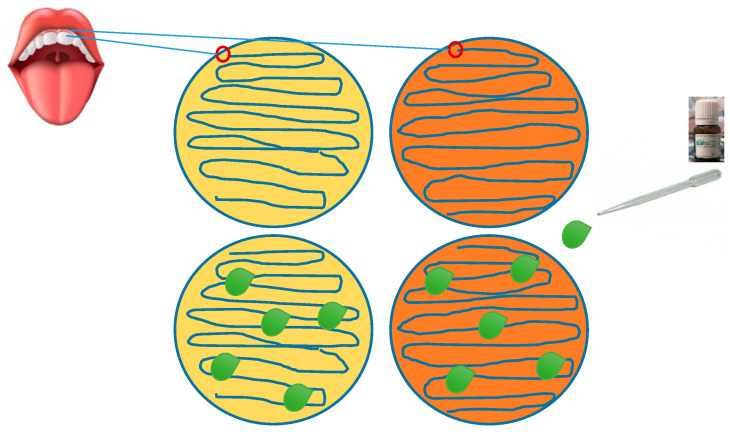
Effect in vitro of Salistat SGL03 on oral microbiota. Yellow dish—universal medium, orange dish—selective medium. Green drops—*L.salivarius* SGL03 included in Salistat SGL03 causing growth retardation zones.

**Figure 2 molecules-25-04519-f002:**
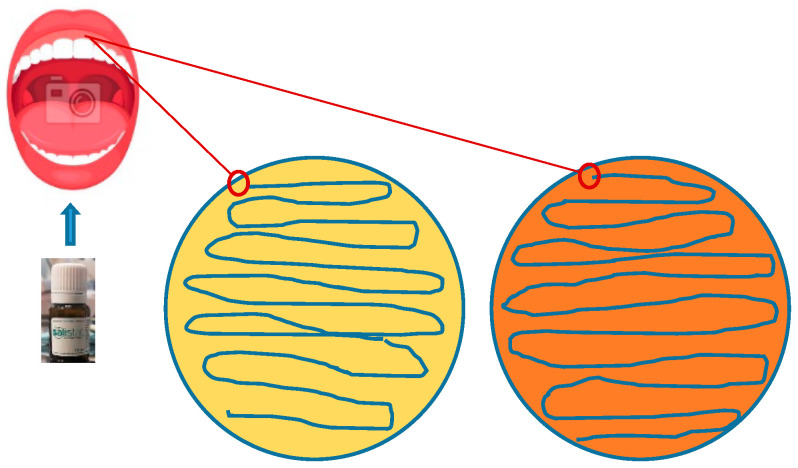
Effect in vivo of Salistat SGL03 on oral microbiota. Yellow dish—universal medium, orange dish—selective medium for *Lactobacillus* species.

**Figure 3 molecules-25-04519-f003:**
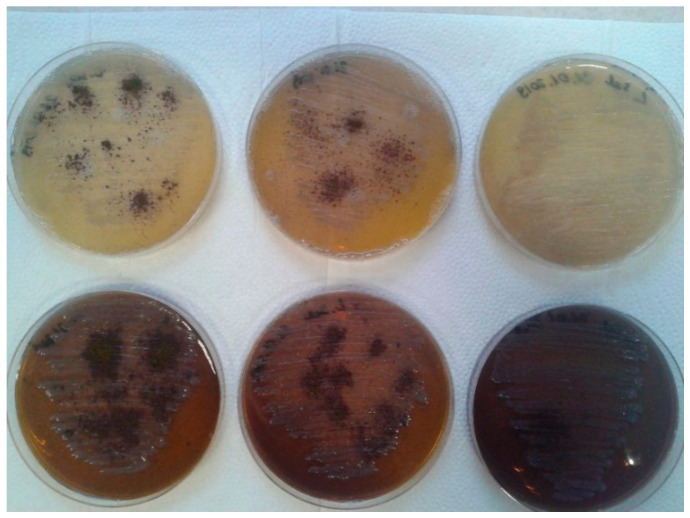
Upper row. 1. *P. gingivalis* control. 2. *P. gingivalis* and Salistat SGL03 treatment after 48 h incubation at 37 °C. 3. *P. gingivalis* and Salistat SGL03 treatment 8 days incubation at 37 °C (complete medium). Lover row 1. *P. gingivalis* control 2. *P. gingivalis* and Salistat SGL03 after 48 h incubation at 37 °C. 3. *P. gingivalis* and Salistat SGL03 treatment after 8 days incubation at 37 °C (incomplete medium).

**Figure 4 molecules-25-04519-f004:**
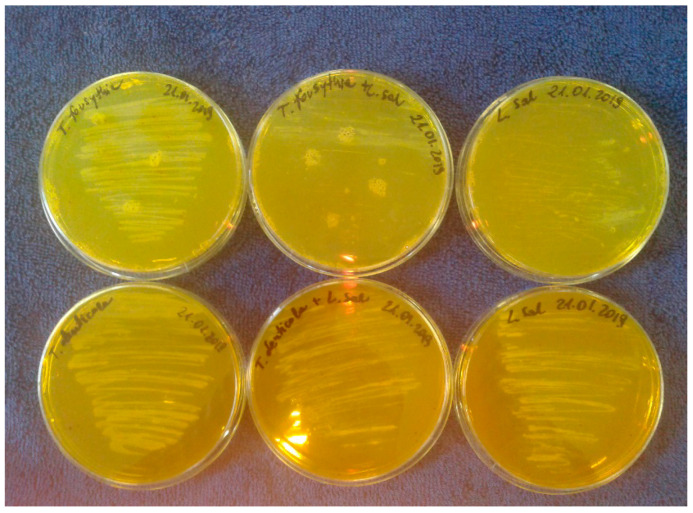
Upper row; 1. *T. forsythia* control. 2. *T. forsythia* and Salistat SGL03 treatment after 24 h incubation. 3. *T. forsythia* and Salistat SGL03 treatment after 48 h incubation at 37 °C. Lower row 1. *T. denticola* control. 2. *T. denticola* and Salistat SGL03 treatment after 24 h incubation at 37°C. 3. *T. denticola* and Salistat SGL03 treatment after 48 h incubation at 37 °C.

**Figure 5 molecules-25-04519-f005:**
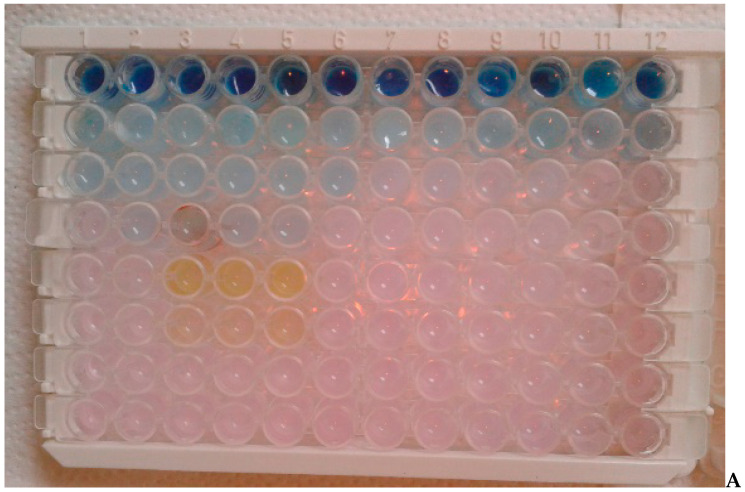
An example of the: MIC analysis of tested bacterial strains (**A**) and MBC analysis of tested bacterial strains (**B**) Lanes from one to 12—bacteria of the red complex with serial dilutions. Lanes 1–4: *P.gingivalis*; Lanes 5–8: *T.denticola*; lanes: 9–12 *T.forsythia*.

**Figure 6 molecules-25-04519-f006:**
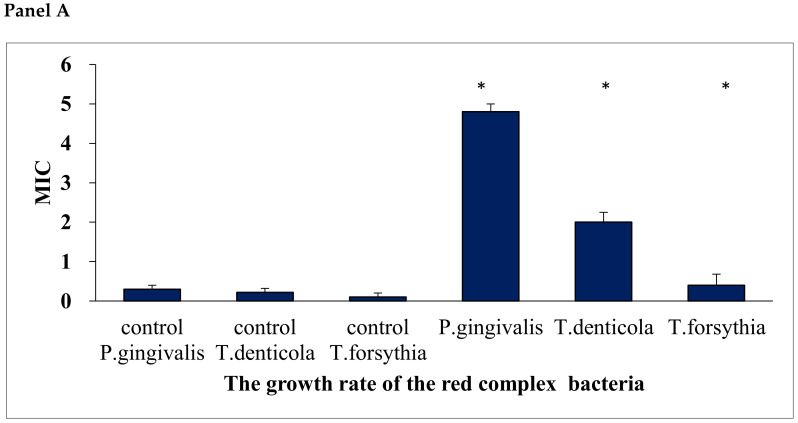
Minimum inhibitory concentration (MIC) for the investigated periopathogens of the red complex (**Panel A**) after the application of Salistat SGL03. Minimum bactericidal concentration (MBC) for the studied red complex periopategens (**Panel B**) after the application of Salistat SGL03. The X axis shows the growth rate analyzed red complex strains. The Y axis represents the MIC (**Panel A**), (MBC) (**Panel B**) value in mM. Statistical significance vs. control at * *p* < 0.05.

**Figure 7 molecules-25-04519-f007:**
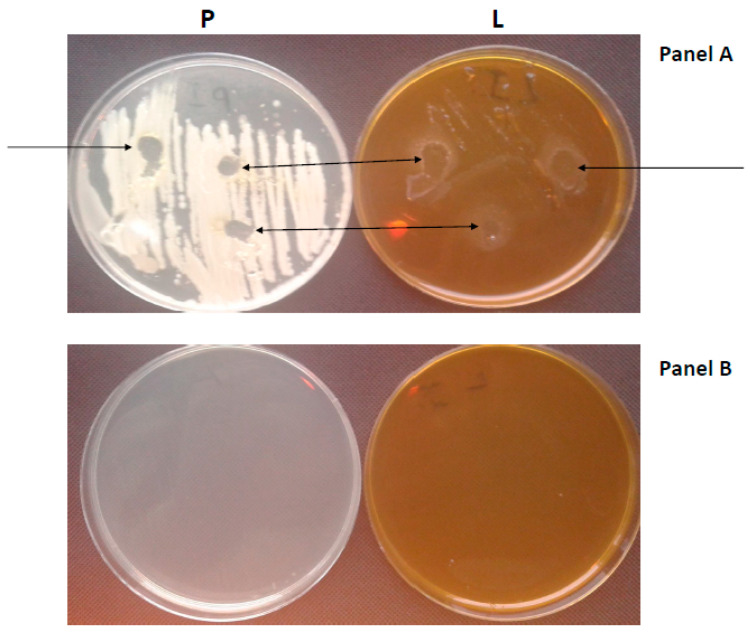
An example of the effect of Salistat SGL03 on bacteria taken from the oral cavity. After 24 h of incubation (pathogens with Salistat SGL03). The zones with growth inhibition are marked with arrows. Lighter (solid) medium marked P and darker (selective) medium marked L. Material (bacterial inoculum) was taken from the oral cavity of female hygienists (female volunteers), seeded and treated in vitro with Salistat SGL03 (**Panel A**). Material after rinsing the mouth (in vivo) with Salistat SGL03 and spreading the bacterial inoculum on the plates (**Panel B**), (details in chapter 2.3 a and b—Materials and Methods and in Appendix A).

**Figure 8 molecules-25-04519-f008:**
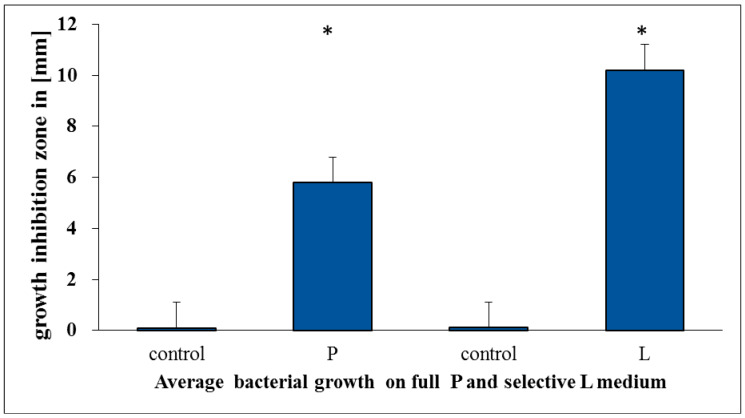
Evaluation of the inhibitory effect of the *L. salivarius* strain present in Salistat SGL03 on selected species of red complex bacteria living in the oral cavity after 24 h of incubation. The growth inhibition zone is expressed in (mm). P—complete medium for the growth of all microorganisms, L—selection medium for the growth of *L. salivarius* (for a detailed description see chapter 2.3 a and b). Statistical significance vs. control at * *p* < 0.05.

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
