# Peer review of "Effect of Ligilactobacillus salivarius and Other Natural Components against Anaerobic Periodontal Bacteria"

_molecules, 2020, doi:10.3390/molecules25194519_

Round 1

Reviewer 1 Report

Authors showed effect of Lactobacillus salivarius and other natural components against anaerobic periodontal bacteria. This is largely the introduction section.  Its length is excessive given its scope. The results and discussion section appears to be complete with respect to results, but seems muddled as far as discussion of the results. Overall, I would suggest extensive revision in combination with re-review for this manuscript. It appears that the English check of the present version has been done by a person not familiar with the contents, and there are numerous sentences that are grammatically correct, but have awkward meaning. I hope that my comment is very useful for the improvement of the article.

Author Response

-The section on the description of periodontal diseases has been shortened and structurally rebuilt.

-The broad scope of the scope results from its subject, we wanted to cover all aspects and characterize the strains of the red complex and L. salivarius itself together with the description and characteristics of the product components to demonstrate the anatgonistic effect of the action of the probiotic strain contained in the product against the natural pathogenic bacterial biofilms formed in the oral cavity.

-All corrections are marked in the text in green.

-The discussion of the results has been revised and completed, also supplemented with new literature citations

-An overall revision of the entire manuscript was performed as suggested by the reviewer. The entire article has been thoroughly rebuilt and checked again in terms of content and language.

-Sentences that were grammatically correct but had a different meaning have been corrected (corrections in the text marked in green)

-Thank you very much for your comment, which was extremely important to us when editing our entire article.

Reviewer 2 Report

The article submitted titled ”Effect of Lactobacillus salivarius and other natural components against anaerobic periodontal bacteria” The author used the commercial product Salistat SGL03 (which contains L. salivarius, lactoferrin,  lemon oil,  rosemary oil, gluco-oligosaccharides and vitamin D) to evaluate the bacteriostatic effect for red-complex by MIC, MBC tests and human experiment. Overall, the author performed a comprehensive study. However the manuscript before publication stills need improvements, and some questions should be clarified.  

  1. To ensure safety, did the author finish the biocompatibility test for Salistat SGL03 product before human experiment, especially in vivo test? For example, cytotoxicity test, skin sensitization test, acute systemic toxicity test, skin Irritation test. Please refer to ISO 10993 International Standard.
  2. The L. salivarius can against dental caries significantly in the past (Curr Oral Health Rep (2017) 4:309318 ). Why author designed the MIC and MBC for periodontal bacteria rather than bacteria of dental caries in this study?
  3. Many sentences are similar to the previous papers, such as BMC Oral Health 2014, 14:110, and  Materials 2020, 13, 2499. Please cite these papers and check plagiarism. In addition, a lot of redundant space or English problems in this version of manuscript. For example: line174, line237: in in? ….   the author should check again.

  Other questions are shown below:

  • -line 185: What kind of the P. g. was the author used? Pg33277, Pg W83 or Pg 381 ? 
  • -line 195: Please cite the Kowalczyk et al 2018 as a number 21, which author already cited in this manuscript  
  • -line 247: It should be “10^8 cfu mL^-1”
  • -line 253: The sentence is incomplete. Also, where is the data in MBC/MIC ratio??
  • -line 204: Why the author only recruited women for this study? 
  • -line 209: How long volunteers didn’t brush their teeth before the experiment? 2 hr, 8 hr or 24 hr? It should be described. 
  • -line 231: According to the Polish law and EU law. I hope author should point out which detail regulation they followed?  
  • -line 289: Where is the Supplementary Material?
  • -line 312: I cannot see the 96 wells in Fig. 3.
  • -line 332 & 335: It should be Fig. 7, not Fig. 8
  • -Figure 5: The caption of this figure is not clear. 
  • -Figure 6: It should be "bacterial growth rate” not “rat”, and the y-axis in Panel B is weird?  
  • -Figure 7: What is panel A and B? It should be described in the caption, even though it already mentioned in line 333~335
  • - Figure 8: The cont and contr are represented “control”? The two words are not consistent. 

Author Response

The article submitted titled ”Effect of Lactobacillus salivarius and other natural components against anaerobic periodontal bacteria” The author used the commercial product Salistat SGL03 (which contains L. salivarius, lactoferrin,  lemon oil,  rosemary oil, gluco-oligosaccharides and vitamin D) to evaluate the bacteriostatic effect for red-complex by MIC, MBC tests and human experiment. Overall, the author performed a comprehensive study. However the manuscript before publication stills need improvements, and some questions should be clarified.  

    1. ensure safety, did the author finish the biocompatibility test for Salistat SGL03 product before human experiment, especially in vivo test? For example, cytotoxicity test, skin sensitization test, acute systemic toxicity test, skin Irritation test. Please refer to ISO 10993 International Standard.

Thank you very much for your accurate suggestions for further research related to the SalistatSGL03 product.

The Salistat SGL03 dietary supplement used is not subject to the regulations for medical devices or cosmetics that require such tests, e.g. for skin irritation or cytotoxicity or toxicity. The product is safe and used as a supplement to oral hygiene for the whole family as well as for pregnant and lactating women. It is intended for people who are preparing for or after orthodontic, prosthetic, surgical and endodontic procedures, after scaling and cleaning the roots of the teeth, with periodontitis. Dietary supplements are subject to the Food and Nutrition Safety Act, so they are part of the diet. In the future, we intend to expand our research with a cytotoxicity test, skin sensitization test, acute systemic toxicity test, and a skin irritation test based on the international standard ISO 10993 which concerns the determination of the effect of medical devices on tissues. Except for EN ISO 10993-2: 2006 and EN ISO 10993-10: 2013, the remaining parts of EN ISO 10993 have the status of harmonized standards with Directive 93/42 / EEC for medical devices and Directive 90/385 / EEC for active medical devices for implantation. The provisions of these directives are implemented by the Act of 20 May 2010 on medical devices (Journal of Laws of 2019, item 175, as amended).

The above description is also included in the manuscript discussion (lines 518-533). All corrections as suggested by the reviewer were included in the manuscript and marked in green.

We also got acquainted with the standard of ISO norm 10993.

  1. The L. salivarius can against dental caries significantly in the past (Curr Oral Health Rep (2017) 4:309–318 ). Why author designed the MIC and MBC for periodontal bacteria rather than bacteria of dental caries in this study?

Mr Allaker's work marked with number 2 was included in the list of references and quoted in response to the reviewer's suggestions also in the manuscript (lines 430-439, corrections in the manuscript are marked in green).

 Responses to the reviewer's suggestions regarding the use of the MIC and MBC along with the relevant literature citations are included in lines 440-443, corrections in the manuscript are marked in green)

  1. salivarius can significantly counteract tooth decay [2] and bacteria inducing periodontal disease - where the anaerobic bacterial biofilms are mostly- Ag. Actinomycetemcomitans, P. gingivalis, T. forsythia, T. denticola, P. intermedia [2] from both regions of the mouth are the same [51]. A consequence of the subgingival colonization by bacteria of the structure surrounding the tooth that is bone, ligament (periodontium), gum and root cementum, are symptoms of oral cavity infection, as microbes can induce an inflammatory response that leads to redness, swelling of the gums, bleeding and eventually tissue degradation and through successive stages of loosening teeth - their recession [2]. In patients with periodontal disease, bacteria also colonize implants. Over time, the bacterial plaques on the surface of the teeth grow larger and descend below the gumline. Subsequently, bacteria and their products from the gingival pockets can pass through the blood vessels to the periphery and contribute to the development of systemic diseases. Therefore, the performance of MIC and MBC tests are an excellent "tool" for the analysis of these bacterial complexes. MIC and MBC are the basic indicators of the effectiveness of targeted antibiotic therapy and bacterial drug susceptibility testing in the chemotherapy of bacterial infections [52].

51.Educational Committee of the International Society of Pediatric Dentistry (USA). Insights into the mouth. Borgis - New Stomatology 4/1999, s. 3-8.

  1. Pietrocola G, Ceci M, Preda F, Poggio C, Colombo. Evaluation of the antibacterial activity of a new ozonized olive oil against oral and periodontal pathogens. J Clin Exp Dent. 2018 Nov; 10(11): e1103–e1108. Published online 2018 Nov 1. doi: 4317/jced.54929

  1. Many sentences are similar to the previous papers, such as BMC Oral Health 2014, 14:110, and  Materials 2020, 13, 2499. Please cite these papers and check plagiarism. In addition, a lot of redundant space or English problems in this version of manuscript. For example: line174, line237: in in? ….   the author should check again.

We tried to correct the quoted sentences so that they do not overlap with the previous articles. However, sometimes the selection of terms and vocabulary is often precisely defined and requires their use in the appropriate context with the appropriate meaning.

We have also inserted appropriate quotations of these articles into the text, line 474 position 57 and line 204 position 58

We have rewritten the entire manuscript in terms of form and English

  Other questions are shown below:

  • -line 185: What kind of the P. g. was the author used? Pg33277, Pg W83 or Pg 381 ?  P.g was 33277 -line 193 in the manuscript
  • -line 195: Please cite the Kowalczyk et al 2018 as a number 21, which author already cited in this manuscript  
  • Was quoted and amended, line 204 of the manuscript
  • -line 247: It should be “10^8 cfu mL^-1”
  • Has been corrected
  • -line 253: The sentence is incomplete. Also, where is the data in MBC/MIC ratio??
  • the sentences were corrected, lines 301-302, results Fig 6 Panel A and B
  • -line 204: Why the author only recruited women for this study? 
  • Dental hygienist is a female profession, in these training groups only women applied for research, while men in this profession are very rarely educated - hence their lack in the analyzed research groups)- this description is given in the text in lines 211-213
  • -line 209: How long volunteers didn’t brush their teeth before the experiment? 2 hr, 8 hr or 24 hr? It should be described.
  • All volunteers after eating their first meal - do not brush their teeth approx. 4 hours before the experiment. Swabs were collected only from starch from the surface of the teeth after consuming a product that was registered in the Sanitary Inspection as a dietary supplement and is sold on the market. The participants of the study (volunteers) were informed about the course of the study and gave their consent. In the first stage, periodontal inoculum (s) were grown in Petri dishes with complete medium marked (P) and selective for Lactobacillus marked (L).- this description is given in the text in lines 221-227
  • -line 231: According to the Polish law and EU law. I hope author should point out which detail regulation they followed?  

In general, it was about the use of the preparation and the consent to collect the material from the teeth from the volunteers themselves, so the approval of the bioethics committee was not needed as there was no invasive interference with the periodontal tissues. Pursuant to Polish law and EU law, consent for such tests is not required. However, the tests were performed in accordance with the legal acts presented below:

  • Act of 5 December 1996 on the profession of a doctor and dentist (Journal of Laws of 1997, No. 28, item 152) (Announcement of the Marshal of the Sejm of the Republic of Poland of February 22, 2019 on the publication of the uniform text of the Act on the professions of doctor and dentist, Journal of Laws 2019, item 537).
  • Act of September 6, 2001, Pharmaceutical Law, Journal of Laws 2001, No. 2001 no.126, item 1381 (Announcement of the Marshal of the Sejm of the Republic of Poland of February 22, 2019 on the publication of the uniform text of the Act - Pharmaceutical Law, Journal of Laws 2019, item 499)
  • Act of September 6, 2001, Pharmaceutical Law (Journal of Laws of 2017, item 2211, as amended) and the Act of May 20, 2010 on medical devices (Journal of Laws of 2017, item 211) as amended)
  • Act of 1 July 2005 on the collection, storage and transplantation of cells, tissues and organs (Journal of Laws of 2005, No. 169, item 1411);
  • Act of 20 May 2010 on medical devices, Journal of Laws 2010, No. 2010 no.107, item. 679 (Announcement of the Marshal of the Sejm of the Republic of Poland of December 13, 2019 on the publication of the consolidated text of the Act on Medical Devices, Journal of Laws 2020, item 186)
  • Regulation of the Minister of Health of 2 May 2012 on Good Clinical Practice (Journal of Laws of 2012, item 489);
  • Regulation of the Minister of Health of 16 February 2016 on detailed requirements for planning, conducting, monitoring and documenting a clinical trial of a medical device, Journal of 2016 item 209
  • Recommendation (2006) 4 of the Committee of Ministers of the Council of Europe for member states on research on biological materials of human origin (Recommendation Rec (2006) 4 of the Committee of Ministers to member 13 states on research on biological materials of human origin), on March 15, 2006;
  • EU Directive 2004/23 / EC of the European Parliament and of the Council of 31 March 2004 on setting standards of quality and safety for the donation, procurement, testing, processing, preservation, storage and distribution of human tissues and cells (OJL102, 7.4 .2004, p. 48).
  • Helsinki Declaration of the World Association of Physicians (WMA) Ethical Principles of Conducting Medical Research with Human Participation -WMA_-October-2013.- this description is given in the text in lines 247-278
  • -line 289: Where is the Supplementary Material?
  • Supplementary material will be attached as a separate file which will contain plates with zones of growth inhibition in volunteers
  • -line 312: I cannot see the 96 wells in Fig. 3.
  • in figure 5 previously as 3 is described in the text 96 wells, line 361 in the manuscript
  • -line 332 & 335: It should be Fig. 7, not Fig. 8
  • has been corrected
  • -Figure 5: The caption of this figure is not clear. 
  • the caption under the drawing has been changed to the correct one
  • -Figure 6: It should be "bacterial growth rate” not “rat”, and the y-axis in Panel B is weird?  
  • the captions in the chart have been corrected
  • -Figure 7: What is panel A and B? It should be described in the caption, even though it already mentioned in line 333~335
  • Panels A and B are now spelled correctly in the text and below the picture
  • - Figure 8: The cont and contr are represented “control”? The two words are not consistent. 
  • The control words in the chart have been corrected

Round 2

Reviewer 1 Report

Accept in present form.